# Monosomy-3 Alters the Expression Profile of the Glucose Transporters GLUT1-3 in Uveal Melanoma

**DOI:** 10.3390/ijms21249345

**Published:** 2020-12-08

**Authors:** Tjorge Maaßen, Siranush Vardanyan, Anton Brosig, Hartmut Merz, Mahdy Ranjbar, Vinodh Kakkassery, Salvatore Grisanti, Aysegül Tura

**Affiliations:** 1Department of Ophthalmology, University of Lübeck, Ratzeburger Allee 160, 23538 Lübeck, Germany; tjorge.maassen@web.de (T.M.); siran1992@mail.ru (S.V.); Anton.Brosig@uksh.de (A.B.); Mahdy.Ranjbar@uksh.de (M.R.); Vinodh.Kakkassery@uksh.de (V.K.); Salvatore.Grisanti@uksh.de (S.G.); 2Reference Center for Lymph Node Pathology and Haematopathology, 23562 Lübeck, Germany; merz@haematopathologie-luebeck.de

**Keywords:** uveal melanoma, monosomy-3, glucose transport, GLUT1, GLUT2, GLUT3

## Abstract

Monosomy-3 in uveal melanoma (UM) cells increases the risk of fatal metastases. The gene encoding the low-affinity glucose transporter GLUT2 resides on chromosome 3q26.2. Here, we analyzed the expression of the glucose transporters GLUT1, GLUT2, and GLUT3 with regard to the histological and clinical factors by performing immunohistochemistry on the primary tumors of *n* = 33 UM patients. UMs with monosomy-3 exhibited a 57% lower immunoreactivity for GLUT2 and a 1.8×-fold higher ratio of GLUT1 to total GLUT1-3. The combined levels of GLUT1-3 proteins were reduced in the irradiated but not the non-irradiated tumors with monosomy-3. GLUT3 expression was stronger in the irradiated samples with disomy-3 versus monosomy-3, but the ratio of the GLUT3 isoform to total GLUT1-3 did not differ with regard to the monosomy-3 status in the irradiated or non-irradiated subgroups. Systemic metastases were associated with the presence of monosomy-3 in the primary and circulating tumor cells as well as a higher GLUT1 ratio. Upregulation of the high-affinity glucose transporter GLUT1 possibly as a compensation for the low-affinity isoform GLUT2 may be enhancing the basal glucose uptake in the UM cells with monosomy-3. Prevention of hyperglycemia might, therefore, be a valuable approach to delay the lethal UM metastases.

## 1. Introduction

Uveal melanoma (UM) is the most frequent primary intraocular malignancy in adults, which gives rise to lethal metastases predominantly in the liver in approximately 50% of the patients regardless of the local tumor control [1,2]. Due to the lack of efficient therapies, UM metastasis sadly results in an average survival time of less than six months following diagnosis [1,2,3]. The most important prognostic factor for UM is the loss of one copy of chromosome 3 (monosomy-3) in the primary tumor cells, which significantly increases the likelihood of systemic metastases in patients having this aberration [4,5,6]. Likewise, an earlier study, which included *n* = 11 UM patients with pathologically confirmed metastatic disease reported that the metastases of UMs with monosomy-3 (*n* = 4 patients, 36.4%) exhibited a more rapid disease progression compared to the tumors with disomy-3 or partial chromosome 3 anomalies (*n* = 7 patients, 63.6%) [7]. However, the current knowledge on the molecular mechanisms underlying the highly aggressive metastatic potential of the UMs with monosomy-3 remains insufficient for the development of preventive therapies.

All rapidly proliferating solid malignancies overgrow their blood supply at a certain point, becoming gradually deprived of oxygen and nourishment. The cancer cells accommodate to such metabolic conditions by mainly executing glycolysis in the cytosol instead of oxidative phosphorylation in the mitochondria even in the presence of oxygen, which is commonly known as the Warburg effect [8,9,10]. The upregulation of glycolysis in tumor cells can enable the faster production of energy and other biosynthetic materials as long as the glucose is abundant and its cellular influx is maintained at high levels [9,11]. This feature of cancer cells is exploited in clinical practice through the administration of a radioactive isoform of glucose (18-fluoro-2-deoxyglucose, FDG) to image the malignant tumors with an elevated metabolic rate by FDG-positron emission tomography (PET) scans [8,9].

Remarkably, the metabolic activity of primary UMs in FDG-PET scans was positively correlated with the presence of monosomy-3 in these tumors [12,13]. These findings, therefore, suggest that the UM cells with monosomy-3 execute a higher rate of glucose influx. However, it is still not known how the uptake of glucose is accomplished in the UM cells. Owing to its hydrophilic nature and large size, glucose cannot diffuse passively across the plasma membrane, but relies on the translocation of special transport proteins onto the cell surface. Currently, there are two known major groups of glucose transporters, which are classified according to their energy-dependence. The sodium-linked glucose transporters use the energy from an electrochemical gradient of sodium to transport glucose against its concentration gradient. In contrast, the glucose transporter (GLUT) family of proteins mediate the energy-independent transport of glucose down its concentration gradient by facilitative diffusion in a bidirectional manner [8,14,15].

The GLUT family comprises 14 known isoforms with GLUT1-4 being the most extensively characterized members [8,16,17]. GLUT1 is a high-affinity glucose transporter with a Km for glucose at around 3–7 mM and a strong protein expression in the erythrocytes, brain microvessels, kidney, and colon of adult humans [8,14,15,17]. The GLUT2 protein is a low-affinity glucose transporter (Km = 17 mM) that is expressed in the liver, pancreatic beta cells, small intestine, and kidney [8,14,15,16,17,18]. Owing to its low-affinity, GLUT2 plays a substantial role in the maintenance of glucose homeostasis by enabling the liver and pancreas to monitor the elevations in blood glucose levels [8,14,16,18]. GLUT3 is a very high-affinity glucose transporter with a Km for glucose at around 1.5 mM [8,14,16], which is considerably lower than the normal fasting blood glucose concentrations of 3.5–5.5 mM [19]. Expression of the GLUT3 protein is confined mainly to the nerve cells of the brain, consistent with the massive glucose demand of this tissue [15,16,17,18]. In contrast, the mRNAs of both GLUT3 and GLUT1 are widely distributed in human tissues [8,14], suggesting that the cell-specific expression of these glucose transporters at the protein level may be regulated by post-transcriptional control mechanisms [20,21]. GLUT1 and GLUT3 also enable the transport of galactose, mannose, xylose, and dehydroascorbic acid with varying affinities [8,14,17]. GLUT4 is another high affinity transporter (Km for glucose around 5 mM), which is mainly present on the muscle and fat cell membranes [8,18]. However, unlike the GLUT1-3 isoforms, which can operate in an insulin-independent manner, the GLUT4 protein is translocated from the internal storage vesicles on to the cell membrane via insulin [8,14,15,16,17]. Depending on their function, different cell types appear to have a combination of multiple GLUT isoforms with distinct kinetic and regulatory features [8,14,16,17,18,21,22,23,24]. The ratio of the individual GLUT isoforms to each other can also vary within the cells of a given tissue depending on the cellular differentiation state [24,25,26,27].

Only the gene encoding the GLUT2 isoform (Gene ID: 6514), which has the lowest affinity for glucose among the known members of the GLUT family, resides on chromosome 3 [14,17,28]. In addition, the GLUT1 protein can become upregulated in the UM cells under hypoxia [29] and the knockdown of the hypoxia-response genes in a mouse xenograft model of UM could suppress the expression of GLUT1 [30]. However, it is not known yet, which glucose transporters predominate in the primary UM samples and whether the presence of monosomy-3 is correlated with the expression of another GLUT isoform with a higher affinity as a possible compensation for GLUT2. To gain the first insight into these issues, we performed an immunohistochemical analysis of the glucose transporters GLUT1, GLUT2, and GLUT3 on the primary tumors of *n* = 33 UM patients operated in our clinic and evaluated the protein expression with regard to the prognostic factors. We did not include the insulin-dependent transporter GLUT4 in this study due to the recently reported association of insulin resistance with a more aggressive course of UM [31]. By focusing on the major insulin-independent isoforms GLUT1-3, we have, therefore, attempted to determine whether the primary UMs exhibit a potential for glucose uptake regardless of the insulin-signaling and how the presence of monosomy-3 would influence this aspect. Our findings provide the first sign for a more favorable constellation of hexose transporters in the primary UMs with monosomy-3 that may be increasing the risk of systemic metastases by enhancing the rate of glucose influx.

## 2. Results

### 2.1. Patient Characteristics

The expression of GLUT1-3 was analyzed by immunohistochemistry in *n* = 34 primary UM samples obtained from *n* = 33 patients. Two samples from different time points (partial resection and enucleation) were available from one patient (female, 77 years). Liver or bone metastases were detected in two patients at the initial presentation (male, 81 years and female, 39 years, respectively) whereas all the remaining patients were diagnosed with primary UM without metastases or extraocular growth. Irradiation was performed at a median of two months (range: 1–23 months, data missing from *n* = 3 patients) prior to the acquisition of *n* = 19 of these samples. The clinical data of the patients and the histological characteristics of their primary tumors are summarized in Table 1.

### 2.2. Expression of GLUT1-3 with Regard to the Monosomy-3 Status of the UM Samples

The intensity of the immunostainings was determined in the images acquired from the entire tumor area using a semiquantitative approach (Figure 1A, please see the methods for a detailed description of the quantification). To verify the specificity of our antibodies, we also performed immuno-blottings under denaturing and non-reducing conditions, which demonstrated that the antibodies we used were highly specific for the respective GLUT isoforms against which they were generated, without any cross reactivity for the other GLUT isoforms tested (Figure 1B).

The immunostainings for GLUT1, GLUT2, and GLUT3 revealed positive signals with intra-tumoral heterogeneity in all of the UM samples (*n* = 34) regardless of the irradiation status. The intensity of GLUT2 was approximately 57% lower in the tumors with monosomy-3 vs. disomy-3 (*p* = 0.0006, Mann-Whitney U test, *n* = 14 and 20 samples, respectively). The intensity of GLUT3 and the total levels of GLUT1-3 were also reduced by approximately 40% in the monosomy-3 tumors, but this effect did not reach significance (*p* = 0.071 and *p* = 0.061, respectively, Mann-Whitney U test). In contrast, the intensity score for GLUT1 did not exhibit any difference with regard to the monosomy-3 status (*p* = 0.641, Mann-Whitney U test, Figure 1C). However, the median ratio of GLUT1 to the total levels of GLUT1-3 was 1.8-fold higher, whereas the median ratio of GLUT2 to the total GLUT1-3 was 17% lower in the monosomy-3 tumors (*p* = 0.003 and *p* = 0.025, respectively, Mann-Whitney U test, Figure 1D). The median ratio of GLUT3 to total GLUT1-3 did not differ significantly with regard to the chromosome 3 status (*p* = 0.545, Mann-Whitney U test, Figure 1D).

Among the non-irradiated samples (*n* = 15), the total intensity score for GLUT1-3 was reduced in the tumors with monosomy-3 (median: 2.76, range: 0.82–5.95) compared to disomy-3 (median intensity: 3.28, range: 0.89–5.23), but this effect did not reach a significance (*p* = 0.892, two-sided *t*-test, *n* = 8 and 7 tumors, respectively, Figure 2A–F). The non-irradiated tumors with monosomy-3 exhibited a mainly epithelioid morphology with an approximately 1.8-fold increase in the ratio of GLUT1 to the total levels of GLUT1-3 (*p* = 0.014, two-sided *t*-test, Figure 2A–C,F). GLUT1 also tended to be more strongly localized to the plasma membrane in the monosomy-3 tumors (Figure 2A). In contrast, the ratio of GLUT2 to total GLUT1-3 was 41% less in the monosomy-3 vs. disomy-3 tumors (*p* = 0.010, two-sided *t*-test, Figure 2A,D). No remarkable difference was observed in the intensity and ratio of GLUT3 with regard to the monosomy-3 status in the non-irradiated tumors (*p* = 0.853 and *p* = 0.912, respectively, Figure 2A,E). The ratios of GLUT1:GLUT2:GLUT3 as calculated from the mean proportions were 1.38:1:1.10 for the monosomy-3 tumors compared to the ratios of 0.51:1:0.68 for the disomy-3 tumors. For all the GLUT isoforms, the intensity of the immunolabeling tended to be usually stronger in the periphery of the tumors compared to the center (Figure 2A).

The immunoreactivity for GLUT1, GLUT2, and GLUT3 also exhibited a heterogenous pattern among the irradiated tumors (*n* = 19, Figure 3A–F). The total levels of these transporters were significantly less in the monosomy-3 tumors (median: 2.46, range: 1.68–4.61, *n* = 6) compared to the disomy-3 samples (median: 4.29, range: 2.94–5.33, *n* = 13, *p* = 0.002, two-sided *t*-test, Figure 3F). Accordingly, the intensities of GLUT2 and GLUT3 were approximately 50% lower in the monosomy-3 tumors (*p* < 0.01 for both two-sided *t*-tests), whereas the expression of GLUT1 was not altered (median: 0.873 vs. 0.829 in the monosomy-3 vs. disomy-3 samples, respectively, *p* = 0.895). The ratio of GLUT1 to the total GLUT1-3 levels was increased by 1.7-fold in the monosomy-3 tumors (*p* = 0.049, two-sided *t*-test, Figure 3C) whereas the ratios of GLUT2 or GLUT3 to total GLUT1-3 did not differ with regard to the chromosome-3 status (Figure 3D,E). The ratios of GLUT1:GLUT2:GLUT3 as calculated from the mean proportions were 1.52:1:1.55 for the monosomy-3 tumors compared to the ratios of 0.78:1:1.26 for the disomy-3 tumors in the irradiated group. The intensity of the immunoreactivity for all the GLUT isoforms was usually stronger in the periphery of the tumors when compared to the center (Figure 3A).

### 2.3. Association of the Metastatic Risk with GLUT1-3 Expression and Prognostic Factors

Analysis of the metastatic risk with regard to the GLUT1-3 expression and other prognostic factors could be performed on *n* = 20 of the 33 patients who had a follow-up time of 2–9 years. Among the excluded patients (*n* = 13), eight patients had a shorter follow-up time whereas the remaining five patients have passed away before we were able to ascertain the absence of systemic metastases. Irradiation was performed in *n* = 12 of the 20 patients (60%) included in the analysis. Systemic metastases were manifested in *n* = 10 of these 20 patients with liver being the primary site in *n* = 9 of the 10 patients. One patient (female, 39 years) has developed metastases only in the bone and brain whereas another patient (female, 69 years) had metastases in the lung subsequent to the liver.

The intensities of GLUT2 and GLUT3 were approximately two-fold higher in the non-metastatic group (*p* = 0.019 for GLUT2, *p* = 0.052 for GLUT3, Mann-Whitney U test), but the ratios of these transporters did not differ remarkably with regard to metastases (Table 2). In contrast, the ratio of GLUT1 was significantly elevated in the metastatic vs. non-metastatic group (41.2% vs. 25.6%, respectively, *p* = 0.023). Likewise, the prevalence of monosomy-3 was higher in both the primary tumors and the circulating melanoma cells (CMC) of the patients who developed metastases (*p* < 0.02, Table 2). The metastatic patients tended to present with the primary tumor in the left eye more frequently (*p* = 0.025, Pearson’s Chi-Square test, *p* = 0.070, Fisher’s exact test). No significant association was observed between the risk of systemic metastases and other histological or clinical parameters such as the total GLUT1-3 levels, age, gender, tumor size, optic nerve invasion, ciliary body invasion, irradiation, and the presence and number of CMC (Table 2).

## 3. Discussion

Despite the well-established association of monosomy-3 with a higher risk of developing lethal UM metastases, the cellular mechanisms underlying this event have not been sufficiently elucidated for the development of efficient and preventive therapies. The extent of monosomy-3 in the primary UMs was positively correlated with the metabolic activity of these tumors in FDG-PET scans [12,13]. This suggests that the malignant UM cells with monosomy-3 exhibit a higher rate of glucose uptake, but the regulation of glucose influx in UM has surprisingly received very little attention so far. Remarkably, the *Slc2a2* gene, which encodes the low-affinity glucose transporter GLUT2, resides on chromosome 3q26.2 [14,17,28]. To gain more insight into the glucose transporter profile of the UM cells, we, therefore, evaluated the expression of the insulin-independent isoforms GLUT1, GLUT2, and GLUT3 in the UM samples of our patients with regard to the monosomy-3 status and clinical factors.

The GLUT isoforms 1–4 exhibit a homology of approximately 48–63% in humans [14]. To exclude the possibility of a cross reaction, we initially verified the specificity of our antibodies by immunoblotting and performed immunohistochemistry for GLUT1-3 on *n* = 34 primary tumors obtained from our UM patients. Here, we report for the first time that the expression of GLUT2 was approximately 57% lower in the monosomy-3 tumors from a mixed cohort of non-irradiated and irradiated samples. In contrast, the ratio of the GLUT1 isoform to total GLUT1-3 was increased by about 1.8× fold in the monosomy-3 tumors of this cohort. An elevated GLUT1 ratio was also associated with a higher risk of developing systemic metastases during the follow-up time of 2–9 years (Table 2). Our findings, therefore, suggest that GLUT1 rather than GLUT3 is the preferentially upregulated isoform in the UMs with monosomy-3 possibly as a compensation for GLUT2.

The major limitations of our study involve the absence of information on the subcellular localization and glycosylation of the GLUT isoforms. Although the expression of GLUT1 tended to be localized more frequently to the plasma membrane in the monosomy-3 tumors, we could not distinguish the exact subcellular expression pattern of the GLUT1-3 isoforms in our study. We were, therefore, not able to estimate the number of transporters that were active on the cellular surface versus stored in intracellular vesicles. The activity of GLUT proteins and their affinity for glucose can be further modulated by the extent of glycosylation [32,33]. The glycosylation of target proteins can influence the binding affinity of the antibodies or the access to their epitopes and, thereby, alter the signal intensities obtained by the immunodetection methods [34,35]. Further limitations arise from the low number of our tumor samples, which originate from a single institute. As the next step, it would be, therefore, highly informative to analyze the expression of all the 14 known GLUT isoforms in normal versus deglycosylated tissue sections as well as the subcellular protein fractions obtained from a larger cohort of tumors from multiple centers.

The combined levels of GLUT1-3 proteins were significantly lower in the irradiated but not the non-irradiated tumors with monosomy-3. A possible reason accounting for this observation might be the downregulation of genes encoding ribosomal components in the UM cells with monosomy-3 [36], which may be hindering the protein translation in the metabolically challenged, irradiated tumors. Accordingly, the intensity of the GLUT3 immunoreactivity was weaker in the irradiated tumors with monosomy-3, but the ratio of the GLUT3 isoform to total GLUT1-3 did not differ in the non-irradiated or the irradiated groups with regard to the monosomy-3 status. The lower immunoreactivity for GLUT3 in the irradiated tumors with monosomy-3 may initially appear as a very contradictory finding due to the stronger affinity of GLUT3 for glucose compared to GLUT1 [15,16,17]. However, as stated earlier, GLUT1 and GLUT3 enable the transport of not only glucose, but also galactose, xylose, mannose, and dehydroascorbic acid with differing affinities. GLUT3 has a two-fold higher affinity for galactose compared to GLUT1, with the Km values of 8.5 mM and 17 mM, respectively [8]. Cells with a higher GLUT3:GLUT1 ratio would, therefore, be expected to take up more galactose, particularly when the extracellular glucose does not exceed the physiological range and interfere with the galactose influx. Remarkably, the utilization of galactose as an alternative energy source could divert the cancer cells from the glycolytic toward the mitochondrial phenotype [37] and induce autophagy-related damage in different tumor cell lines [38]. Likewise, mannose could impair the tumor glycolysis and enhance chemosensitivity in a cell culture and animal studies [39]. Although we could not find direct information on the Km of GLUT3 for mannose, photoaffinity labeling studies have shown that the affinity of bis-mannose for GLUT3 exceeded that of GLUT1 about two-fold [40]. The more efficient uptake of such additional hexoses like galactose and mannose via GLUT3 may, therefore, counter the proliferative effects of the advanced glucose influx through this transporter and be directing the irradiated UM cells with disomy-3 toward a less glycolytic, more quiescent phenotype that deserves further investigation.

In our study, we could not evaluate the molecular mechanisms that mediate the differential expression of the GLUT1 and GLUT2 isoforms in the UMs with monosomy-3. A likely candidate involved in this process might be the transcriptional coactivator YAP1, which could stimulate the mRNA expression of GLUT1-3 in diverse cell types [41,42]. Since an aberrant YAP1 activity is also implicated in the pathogenesis of UM [43,44,45], it would be very informative to conduct further studies on the involvement of YAP1 in the regulation of GLUT isoforms in UM. Additionally, the p53 tumor suppressor protein exerts a negative influence on the expression of GLUT1 via NF-κB [46,47]. Several “hot-spot” mutations of p53 that are frequently observed in various tumors have resulted in the upregulation of GLUT1. However, such mutations have led to a significant increase in the levels of the p53 protein [47]. Remarkably, the overexpression of p53 in primary UMs was also associated with unfavorable factors such as the extraocular extension, vascular invasion, proliferative activity, and an epithelioid morphology [48,49,50]. Nevertheless, the UM cells were found to be exhibiting functional defects in the signaling events downstream of the p53-pathway [51,52], which remain to be elucidated in more detail. The dysregulation of GLUT1 dynamics in response to the p53-overexpression in UM cells may, therefore, be a novel aspect involved in the aggressive course of this disease, which deserves further investigation.

The expression of the GLUT1-3 isoforms in our UM samples tended to be stronger in the tumor periphery when compared to the center (Figure 2A and Figure 3A). This pattern was consistent with the findings of a recent study on the spatial heterogeneity of UM, which demonstrated the higher expression of epigenetic marker proteins in the peripheral versus central tumor cells, likely due to the more intensive exposure of the former cell type to the tumor microenvironment [53]. Interestingly, the environmental conditions, such as the presence of certain growth factors, can also influence the expression of GLUT proteins [46]. For instance, the insulin-like growth factor-1 (IGF1) could promote both the expression and membrane recruitment of GLUT1 and GLUT3 in different cell types [54,55]. IGF1 is considered to be the major chemoattractant responsible for the liver tropism of dissociated UM cells [56] since this hormone is mainly released from the hepatocytes [57] and the upregulation of the IGF1-receptor in UM cells significantly increases the metastatic risk [56]. As its name suggests, IGF1 also exerts an insulin-mimetic activity and, accordingly, becomes more bioavailable under insulin resistance, which is a condition defined by the requirement of higher levels of insulin to maintain the blood glucose levels in the physiological range [57]. Remarkably, insulin resistance was already found to be correlated with a worse prognosis in the patients with UM or choroidal nevi [31]. We have also recently reported a more insulin-resistant gene expression profile in the UMs with monosomy-3, which might interfere with the storage of excessive glucose as normal glycogen in such tumors [58]. However, insulin resistance is not recognized as an established prognostic factor for UM yet and the molecular mechanisms of its association with an adverse prognosis deserve further investigation.

The liver plays a fundamental role in the maintenance of glucose homeostasis by serving as the major glucose storage site in the body. The liver responds to the insulin stimulation by terminating the release of glucose into the bloodstream [59], whereas sustained hyperglycemia is one of the factors that impairs the biological actions of insulin [60]. The insulin-resistant UM patients in the previously mentioned study exhibited a slight but significant elevation in the levels of fasting plasma glucose compared to the patients with choroidal nevi or insulin-sensitive, age-matched controls. However, the plasma glucose levels remained in the “normoglycemic range” in all groups [31], suggesting that a subtle form of hyperglycemia exists in the UM patients, which would remain undetected for many years unless regular blood sample examinations are performed. This mild hyperglycemia may, therefore, accompany the transformation of choroidal nevi to more aggressive melanomas.

Hyperglycemia can further interfere with the normal signaling events by increasing the non-enzymatic glycosylation of cell surface proteins [61], but its impact on the glycation and function of GLUT isoforms or the pathogenesis of UM metastases is not known yet. The only indirect information we could retrieve on the latter subject was from the case report of a 17-year-old male patient with multiple liver metastases of UM. The blood glucose concentration of this patient was measured as 125 mg/dL [62], which corresponds to 6.9 mM and exceeds the normal fasting glucose levels of 3.5–5.5 mM [19], but it was not stated whether the blood sample was collected after fasting [62]. One of the patients in our study (male, 52 years) also had an elevated fasting blood glucose level of 166 mg/dL (9.2 mM) five months after the manifestation of liver metastases, but we could not collect the information on this parameter from the remaining patients. Since the uptake of glucose, which is available at an extracellular concentration of 9.2 mM, can be efficiently mediated by GLUT1 (Km: 3–7 mM) but not GLUT2 (Km: 17 mM), the disseminated UM cells with monosomy-3 and a higher GLUT1:GLUT2 ratio might have profited more from such a hyperglycemic environment. The PET-scans of the liver metastases of UM patients also indicate variations in glucose dynamics during the disease course with a decreased FDG uptake despite the absence of size changes under chemotherapy and an increased FDG influx in the lesions that grow during the follow-up [63]. We, therefore, believe that the impact of hyperglycemia on the pathophysiology of UM urgently deserves further investigation to evaluate the therapeutic potential of targeting an easily modifiable factor like the elevated blood glucose levels.

Further therapeutic approaches can involve the inhibition of GLUT1 by different mechanisms. For instance, the advanced uptake of dehydroascorbic acid, which competes with glucose for the transport through GLUT1 and GLUT3, leads to the accumulation of reactive oxygen species and may consequently induce cell death [46]. The small molecules WZB117 and STF-31, which act as the inhibitors of GLUT1, could also suppress the tumor growth in cell lines and animal experiments while exerting a synergism with the chemotherapeutics such as Cisplatin or Paclitaxel [64,65]. Furthermore, the molecule Fasentin, which was originally known as a stimulator of the FAS-mediated cell death, could specifically inhibit GLUT1 [66]. As an indirect inhibitor, the flavonoid Apigenin limits the synthesis of GLUT1 at both the mRNA and protein levels via the PI3K/Akt signaling pathway [67]. Another flavonoid called Genistein, which is included in herbal products like soy, was capable of suppressing the uptake of hexoses through GLUT1 by acting as a tyrosine kinase inhibitor [68]. Moreover, the antibodies against GLUT1 could arrest the cell growth and induce apoptosis while enhancing the antiproliferative effects of chemotherapeutics in vitro [69]. These potential inhibitors of GLUT1 need a comprehensive exploration of their suitability in clinical practice. The ubiquitous expression of GLUT1 in the human body [70] complicates the targeted administration of such drugs and might necessitate an intraocular application. Nevertheless, further research on the molecular therapeutics against GLUT1 can potentially enable an improvement in the devastating prognosis of UM.

## 4. Materials and Methods

### 4.1. Patient Selection

This retrospective study included *n* = 33 consecutive patients who were diagnosed with UM at the Department of Ophthalmology, University of Lübeck, Germany, between December 2009 and January 2018. Diagnosis of UM was confirmed with clinical and ultrasound examinations performed by a specialized ophthalmologist. The study was approved by the local ethic committee of the University of Lübeck (File number: 10–200, Date: 17 December 2010) and performed in accordance with the guidelines of the Declaration of Helsinki of 1975, revised in 2013. All patients gave informed consent prior to their inclusion into the study.

Standardized A and B scans (I3 eyecubed System-ABD, Ellex Inc., Sacramento, CA, USA) and ultrasound biomicroscopy (VuMax II, Sonomed Inc., Lake Success, NY, USA) were performed to determine the size of the tumor, exact intraocular localization, and ciliary body involvement. Metastatic status of the patients was evaluated by liver function tests (alkaline phosphatase, aspartate aminotransferase, alanine aminotransferase, bilirubin), ultrasound of the abdomen, and the computer tomography of the chest and abdomen.

### 4.2. Immunohistochemistry

Samples of the primary tumor (*n* = 34) were obtained from *n* = 33 UM patients after biopsy, resection, or enucleation, fixed in 4.5% formalin, embedded in paraffin, and processed as 6-μm sections that were collected onto Superfrost Ultra-Plus slides. Deparaffinization was performed in three changes of xylol and two changes of absolute ethanol followed by a graded series of ethanol from 90% to 50% for 5 min each and a final wash in triple distilled water for 5 min. For antigen retrieval, sections were incubated in pre-heated 10 mM sodium citrate buffer and pH 6.0 with 0.025% Tween 20 for 20 min in a steam-cooker. After cooling to room temperature, the slides were rinsed three times for 10 min in phosphate-buffered saline (PBS) and incubated with sterile-filtered blocking buffer (3% bovine serum albumin in 10 mM Tris-HCl, pH 7.5, 120 mM KCl, 20 mM NaCl, 5 mM ethylenediaminetetraacetic acid, and 0.1% Triton X-100) supplemented with 5% goat serum for 30 min at room temperature. Sections were then incubated with the polyclonal rabbit primary antibodies against GLUT1 (Abcam, Cambridge, UK, ab32551, 1:150 dilution in blocking buffer), GLUT2 (Abcam, ab95256, 1:40 dilution), or GLUT3 (Abcam, ab15311, 1:50 dilution) overnight at 4 °C. The negative controls were incubated with the blocking buffer alone. The sections were rinsed three times for 5 min with PBS, incubated in freshly prepared 3% hydrogen peroxide solution in PBS for 15 min to block the endogenous peroxidases, rinsed twice with PBS, and incubated with horseradish peroxidase (HRP)-conjugated goat anti rabbit secondary antibodies (Jackson Immunoresearch, Cambridgeshire, UK, 111-035-003, 1.6 μg/mL in blocking buffer) for 1 h. After three rinses for 5 min with PBS, sections were incubated for 10 min with the HRP green substrate that was freshly prepared by mixing 90 μL of HRP-Green chromogen with 1 mL of HRP-Green buffer as instructed by the manufacturer (42 Life Sciences, Bremerhaven, Germany) and washed for 5 min in triple-distilled water. Nuclear counterstaining was performed with nuclear fast red for 10 min. Sections were then briefly rinsed in triple-distilled water for 1 min, dehydrated for 30 s each in a graded series of ethanol (75–96–2 × 100%), which was followed by two times for 5 min in xylol, and mounted in non-aqueous medium.

For each patient, images of the entire tumor area were acquired under 200× magnification by light microscopy (Leica, Wetzlar, Germany). The semi-quantitative scoring of the immunostainings on these images was performed by an observer who was blinded to the clinical parameters of the patients (T.M.). The scoring method was adapted from earlier procedures that were based on the intensity of the immunoreaction and the area occupied by the cells that exhibit a certain degree of immunolabeling [53,71,72]. The intensity of the immunohistochemical reactions was graded as “0” for the absence of immunoreactivity, “1” for weak, “2” for intermediate, and “3” for strong immunoreactivity (Figure 1A). The tumor area was circumscribed on each image and the integrated density (area × intensity) of the selected region was determined. The sum of the integrated densities was then divided by the total area to determine the mean staining intensity for each tumor. The total number of images quantified for GLUT1, GLUT2, and GLUT3 were *n* = 1738, *n* = 2046, and *n* = 1676, respectively.

### 4.3. Immunomagnetic Enrichment of Circulating Melanoma Cells (CMC)

CMC were isolated from the peripheral blood within three hours of collection as described [73]. Cytospins of the isolated cells were stored at −20 °C until use.

### 4.4. Immuno-Fluorescent In Situ Hybridization (Immuno-FISH)

Frozen cytospins that were aged for 24 h at room temperature were processed for Immuno-FISH as described [74], using the primary mouse anti-melanoma associated chondroitin sulfate proteoglycan (Abcam, ab78284, 1/5 in the immunistochemistry-blocking buffer), anti-NKI/C3 (Abcam, ab213090, 1/20 dilution in blocking buffer), anti-NKI/beteb (Abcam; ab63297, 1/5 dilution in blocking buffer), or polyclonal rabbit anti-Melan-A (Abcam, ab210546, 1:100 dilution in blocking buffer) antibodies, followed by the Alexa 488-conjugated goat anti-mouse (ThermoFisher, Waltham, MA, USA, A-11001, 1:100 in blocking buffer) or anti-rabbit secondary antibodies (Abcam, ab150077, 1:100 in blocking buffer). Incubation with the secondary antibodies and all the subsequent steps were performed under protection from light. The chromosome enumeration probe CEP3 (Abbott GmbH, Wiesbaden, Germany, 06J36-003) was applied on to the cells at a volume of 4 μL/cytospin. Nuclei were counterstained with 0.5 μg/mL DAPI in PBS for 10 min. Cells were mounted in Mowiol and analyzed by fluorescence microscopy (Leica DMI 6000 B) using the relevant filter sets (A4: Excitation (Ex): 360/40, Emission (Em): 470/40 nm; L5: Ex: 460/40, Em: 527/30 nm, Cy3: Ex: 545/30, Em: 610/75 nm). Images were acquired using a monochrome digital camera (DFC 350 FX, Leica) and the Leica Application Software (Advanced Fluorescence 2.3.0, build 5131).

For the Immuno-FISH analysis on the paraffin sections of primary tumor samples, deparaffinized sections were processed for antigen retrieval and incubated in blocking buffer as described above for immunohistochemistry, which was followed by the overnight incubation with polyclonal rabbit antibodies against Melan-A (1:100 in blocking buffer) at 4 °C and Alexa 488-conjugated goat anti-rabbit secondary antibodies (1:100 in blocking buffer) for 1 h under protection from light. FISH was then performed as described in Reference [65] with the extension of hybridization to 72 h and using 10-μL probe/cm^2^.

### 4.5. Assessment of the Monosomy-3 Status

The copy number of chromosome 3 in the tumor samples and CMC was determined in the overlay images of cells positive for the melanoma markers. FISH-signals on the retina of *n* = 3 patients were also quantified as a positive control for diploidy. Monosomy-3 was defined by the presence of only one distinct signal or a doublet signal (two signals that were separated by a distance of less than twice the diameter of a signal) in the nucleus. Signal quantification was performed by determining both the percentage of the cells with monosomy-3 within an area and the chromosomal index as described [75]. For the former method, the percentage of cells with or without monosomy-3 within a given area was calculated in a minimum of *n* = 203 non-overlapping nuclei except for one patient with a very small sample (*n* = 88 quantified nuclei). The chromosomal index was determined by counting the total signals for chromosome 3 and dividing by the number of nuclei in a given area. The median values of both quantifications were calculated. Tumors with a percentage of monosomy-3-positive cells equal to or above the median were scored as “1”. The score “1” was also given to the tumors with a chromosomal index that was equal to or below the median. The tumors that received the score “1” for both parameters were classified as monosomy-3 tumors whereas the samples that received the score “0” with both methods were classified as disomy-3. Tumors that received the score “1” with only one method were classified as intermediate. For the statistical analysis, the low and intermediate groups were collectively defined as “disomy-3 tumors”.

### 4.6. Immunoblotting

Electrophoresis was performed using 4–10% TGX stain-free polyacrylamide gels (Bio-Rad, Munich, Germany) under denaturing and non-reducing conditions using the GLUT1 peptide (Abcam, ab33006, 5 μg/well), GLUT2 peptide (Abcam, ab105630, 5 μg/well), GLUT3 peptide (Abcam, ab105625, 5 μg/well), or full-length recombinant human GLUT3 protein (Abcam, ab152681, 200 ng/well). To control for protein loading, gels were visualized using the ChemiDoc MP stain-free imaging (Bio-Rad) by placing the gels directly on the imaging tray with 2–3 mL of distilled water underneath and activating with ultraviolet-light for 5 min. The gels were then equilibrated in blotting buffer (48 mM Tris, 39 mM Glycine, 40% methanol (*v*/*v*)) for 5 min and transferred onto methanol-activated polyvinylidene difluoride membranes by semi-dry blotting (Biotec-Fischer, Reiskirchen, Germany) at a constant current of 0.8 mA/cm^2^ for 30 min. Membranes were blocked in 5% nonfat dry milk in Tris-buffered saline with 0.1% Tween-20 (TBST) for 1 h under gentle agitation, which was followed by incubation with rabbit primary antibodies against GLUT1 (1:500), GLUT2 (1:2000), or GLUT3 (1:1000) diluted in blocking buffer overnight at 4 °C, and the StarBright Blue 700 goat anti-rabbit secondary antibodies (Bio-Rad, 12004161, Ex_max_/Em_max_ = 470 nm/700 nm, 1:1000 in blocking buffer) for 1 h at room temperature. The membranes were washed six times for 5 min in TBST and the detection of fluorescent signals was performed by using the ChemiDoc MP system. The gray values of the fluorescent blots were inverted to obtain dark signals against a clear background.

### 4.7. Statistical Analysis

Data were analyzed using the NCSS statistical software (Version 8.0.13, NCSS, LLC, Kaysville, UT, USA) under Windows 10. The Mann-Whitney U test was performed to analyze the association of numerical parameters with binary variables for the groups with a minimum size of *n* = 10 samples. For groups with smaller sample sizes, an unpaired, two-sided *t*-test was applied assuming equal variance. The proportions of categorical variables were evaluated by Pearson’s Chi-square and Fisher’s exact tests. *p* values less than 0.05 were taken as significant.

## 5. Conclusions

In conclusion, our findings provide the first insight into the monosomy-3-dependent alterations in the glucose transporter profile of UM cells. Upregulation of the high-affinity transporter GLUT1 possibly as a compensation for the low-affinity isoform GLUT2 may be a novel mechanism that increases the metastatic potential of the UM cells with monosomy-3 by enhancing the basal level of glucose uptake. Avoiding hyperglycemia and counteracting insulin resistance to prevent the excessive glucose release from the liver might, therefore, be an affordable and immediately available therapeutic strategy to delay the growth of the UM cells that were released into the circulation or have formed micrometastases in the liver, which deserves further investigation. Future studies involving the full spectrum of the known GLUT isoforms with regard to their subcellular localization and extent of glycosylation on the samples from a larger cohort of UM patients with a longer follow-up would also provide a valuable insight into the neglected field of glucose transport in the UM cells, which may, in turn, enable the identification of novel therapeutic targets.

## Figures and Tables

**Figure 1 ijms-21-09345-f001:**
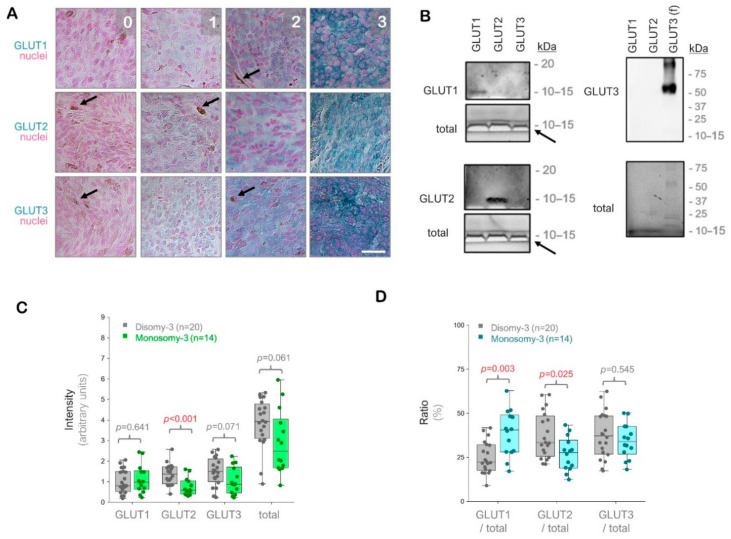
Analysis of the glucose transporters GLUT1-3 in the primary uveal melanomas (UMs) by immunohistochemistry. (**A**) Representative images from the immunostainings for GLUT1-3. Signal detection was performed using the horseradish peroxidase-green substrate, which generates a blue-green reaction product. The nuclei were counterstained with nuclear fast red. The strength of the immunohistochemical reaction within a circumscribed area was graded using a scale of 0–3 (0: negative. 1: weak. 2: intermediate. 3: strong). The mean intensity of the entire tumor area was determined by a semiquantitative approach as described in the Methods section. Images were acquired at an original magnification of 200×. Arrows indicate the pigmented regions. Scale bar = 25 μm. (**B**) Immunoblots with GLUT1-3 peptides or full-length recombinant human GLUT3 protein (f), demonstrating the specificity of the antibodies used. The total protein amount in the wells was detected by the stain-free imaging of gels after ultraviolet-activation. The white bands (arrows) correspond to the bromophenol blue dye in the loading buffer, which has not migrated out of the gels. kDa: Kilodalton. (**C**) Quantification of the GLUT1, GLUT2, and GLUT3 intensities in the primary UMs with regard to the monosomy-3 status. The sum of the GLUT1-3 intensities was also presented as “total”. *p*-values were determined by the Mann-Whitney U test. (**D**) Ratios of GLUT1, GLUT2, and GLUT3 to the total GLUT1-3 levels in the primary UMs with respect to the monosomy-3 status. *p*-values were analyzed by the Mann-Whitney U test. The significant *p*-values (<0.05) in (**C**,**D**) were stated in red.

**Figure 2 ijms-21-09345-f002:**
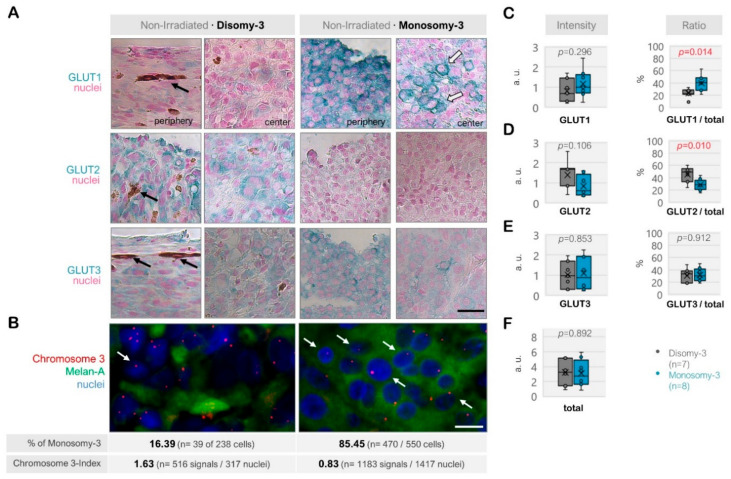
Expression of the glucose transporters GLUT1, GLUT2, and GLUT3 in the non-irradiated uveal melanoma (UM) samples. (**A**) Representative images from the immunostainings for GLUT1-3. The disomy-3 tumor was obtained from a male patient (69 years), who did not develop metastases during the follow-up, whereas the monosomy-3 tumor was removed from a female patient (39 years), who presented with bone metastases that have subsequently spread to the brain. Images were acquired at an original magnification of 200×. The black arrows indicate the pigmentation whereas the white arrows with a dark contour point out the staining of the plasma membrane. Scale bar = 25 μm. (**B**) Immuno-Fluorescent in situ hybridization (Immuno-FISH) was performed to detect the copy number of chromosome-3 (red) in the UM cells that were positive for the melanoma marker Melan-A (green). The nuclei were counterstained in blue with 4′,6-Diamidine-2′-phenylindole dihydrochloride (DAPI). The chromosome 3 status of the tumors was evaluated by determining the percentage (%) of cells with monosomy-3 and the chromosome 3-index as described in the Methods section. The values obtained for these parameters in the previously mentioned tumors are stated underneath the respective image panels (*n*: number). Arrows indicate several examples of cells with monosomy-3. Scale bar = 10 μm. Quantification of the intensity and ratio of (**C**) GLUT1, (**D**) GLUT2, and (**E**) GLUT3 in the primary UMs with regard to the monosomy-3 status. (**F**) The sum of the GLUT1-3 intensities was also presented as the “total.” *p*-values were determined by the unpaired, two-sided *t*-test. The significant *p*-values (<0.05) were indicated in red. a.u.: Arbitrary units.

**Figure 3 ijms-21-09345-f003:**
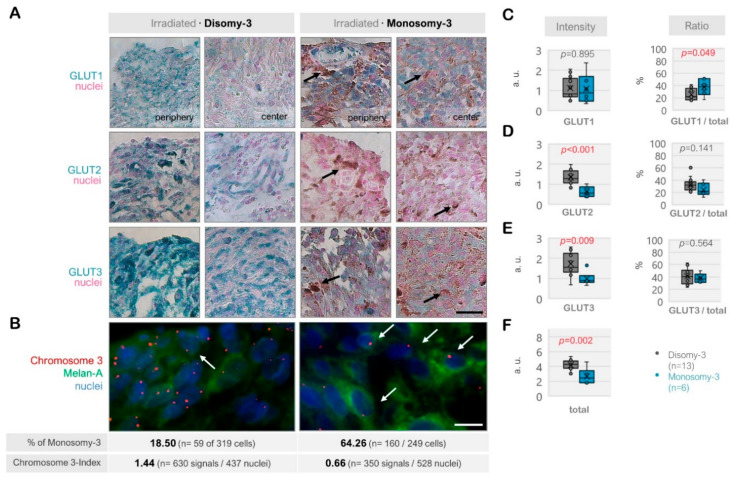
Expression of the glucose transporters GLUT1, GLUT2, and GLUT3 in the irradiated uveal melanoma (UM) samples. (**A**) Representative images from the immunohistochemistry for GLUT1-3. The disomy-3 tumor originates from a male patient (60 years), who remained metastasis-free during the follow-up, whereas the monosomy-3 tumor was obtained from another male patient (52 years), who has developed liver metastases. Irradiation was performed two months prior to the acquisition of both tumors. Images were collected at an original magnification of 200×. Arrows indicate varying degrees of pigmentation. Scale bar = 25 μm. (**B**) Immuno-Fluorescent in situ hybridization (Immuno-FISH) analysis, demonstrating the copy number of chromosome-3 (red) in the UM cells that were expressing the melanoma marker Melan-A (green). The nuclei were counterstained in blue with DAPI. Arrows indicate several cells with monosomy-3. The percentage (%) of monosomy-3 cells and the chromosome index of these tumors were presented underneath the respective images (please see the Methods for the detailed description of the calculation of these parameters, *n*: number). Scale bar = 10 μm. Quantification of the intensity and ratio of (**C**) GLUT1, (**D**) GLUT2, and (**E**) GLUT3 in the irradiated primary UMs with regard to the monosomy-3 status. (**F**) The sum of the GLUT1-3 intensities was presented as “total.” *p*-values were determined by the unpaired, two-sided *t*-test. The significant *p*-values (<0.05) were highlighted in red. a.u.: Arbitrary units.

**Table 1 ijms-21-09345-t001:** Baseline characteristics of the patients and tumor samples.

**Patients**	***n* = 33**
Gender, No. (%)	
Female	15 (45.45)
Male	18 (54.55)
Age in years, mean (SD)	66.91 (11.81)
Tumor location, No. (%)	
Ciliary body	1 (3.03)
Choroid	32 (96.97)
Tumor size in mm, mean (SD)	
Base 1	12.24 (4.23) *
Base 2	12.34 (4.74) *
Prominence	7.47 (4.30) *
**Tumor samples**	***n* = 34**
Sample acquisition, No. (%) *	
Biopsy	2 (6.06)
Resection	12 (36.36)
Enucleation	19 (57.58)
Irradiation, No. (%)	
No	15 (44.12)
Yes	19 (55.88)
Prominent cell type, No. (%)	
Spindle	2 (5.88)
Mixed/spindle	3 (8.82)
Mixed	18 (52.94)
Mixed/epithelioid	3 (8.82)
epithelioid	8 (23.53)
Monosomy-3 positive cells, %	
Median	27.74
Min-Max	7.48–85.46
Chromosome 3 index	
Median	1.08
Min-Max	0.65–1.82
Chromosome 3 status, No. (%)	
Disomy-3	12 (35.29)
Intermediate	8 (23.53)
Monosomy-3	14 (41.18)
Chromosome 3 status in	
non-irradiated tumors, No. (%)	15
Disomy-3	4 (26.67)
Intermediate	3 (20.00)
Monosomy-3	8 (53.33)
Chromosome 3 status	
in irradiated tumors, No. (%)	19
Disomy-3	8 (42.11)
Intermediate	5 (26.32)
Monosomy-3	6 (31.58)

No.: Number. SD: Standard deviation. * Data missing from one patient.

**Table 2 ijms-21-09345-t002:** Association of systemic uveal melanoma (UM) metastases with clinical parameters and the levels of the glucose transporters GLUT1-3 (Follow-up time: 2–9 years).

	All Patients	No Metastases	Metastases	*p* *
	(*n* = 20)	(*n* = 10)	(*n* = 10)	
Age at diagnosis				
Median (Min, Max)	67 (39–83)	67 (53–82)	61.5 (39–83)	0.405
Gender, No. (%)				
Female	10 (50.0)	5 (50.0)	5 (50.0)	1.000
Male	10 (50.0)	5 (50.0)	5 (50.0)	(1.000)
Eye, No. (%)				
Right	9 (45.0)	7 (70.0)	2 (20.0)	**0.025**
Left	11 (55.0)	3 (30.0)	8 (80.0)	(0.070)
Irradiation, No. (%)				
No	8 (40.0)	4 (40.0)	4 (40.0)	1.000
Yes	12 (60.0)	6 (60.0)	6 (60.0)	(1.000)
Tumor size in mm Median (Min, Max)				
Base 1	12.5 (1.4–18.3)	11.1 (1.4–18.3)	13.5 (6.5–17.5)	0.143
Base 2	12.1 (1.4–21.8)	9.1 (1.4–21.8)	14.4 (5.6–19.0)	0.190
Prominence	7.4 (0.5–13.5)	7.1 (0.5–12.3)	7.4 (1.3–13.5)	0.290
Optic nerve invasion, No. (%)				
No	19 (95.0)	10 (100)	9 (90.0)	0.305
Yes	1 (5.0)	0 (0)	1 (10.0)	(1.000)
Ciliary body invasion, No. (%)				
No	16 (80.0)	8 (80.0)	8 (80.0)	1.000
Yes	4 (20.0)	2 (20.0)	2 (20.0)	(1.000)
Circulating melanoma cells (CMC)-positive, No. (%)				
No	1 (5.0)	1 (10.0)	0 (0)	0.305
Yes	19 (95.0)	9 (90.0)	10 (100)	(1.000)
CMC-No./50 mL blood				
Median	9.3	8.2	11.0	0.570
(Min, Max)	(0–51)	(0–51)	(3.8–19.5)	
Monosomy-3 in CMC, No. (%) **				
No	5 (26.3)	5 (50.0)	0 (0)	**0.014**
Yes	14 (73.7)	5 (50.0)	9 (100)	**(0.033)**
% of CMC with Monosomy-3				
Median	30.4	6.3	37.5	**0.011**
(Min, Max)	(0–100)	(0–60)	(11.8–100)	
Monosomy-3 in primary tumor, No. (%)				
No	8 (40.0)	8 (80.0)	0 (0)	**0.000**
Yes	12 (60.0)	2 (20.0)	10 (100)	**(0.007)**
GLUT1-GLUT2-GLUT3 intensity Median (Min, Max)				
GLUT1	1.0 (0.2–2.4)	0.9 (0.2–2.1)	1.0 (0.2–2.4)	0.853
GLUT2	0.8 (0.3–1.6)	1.1 (0.4–1.6)	0.6 (0.3–1.6)	**0.019**
GLUT3	1.2 (0.2–2.6)	1.8 (0.3–2.6)	0.9 (0.2–1.9)	0.052
GLUT1 + GLUT2 + GLUT3	3.5 (0.8–6.0)	3.9 (0.9–5.3)	2.5 (0.8–6.0)	0.165
GLUT1-GLUT2-GLUT3 ratio (%) Median (Min, Max)				
GLUT1/total	33.3 (16.2–51.8)	25.6 (16.2–40.3)	41.2 (17.1–51.8)	**0.023**
GLUT2/total	28.1 (12.4–45.2)	30.2 (21.3–45.2)	24.6 (12.4–43.4)	0.280
GLUT3/total	36.1 (22.1–62.5)	44.3 (22.1–62.5)	33.9 (23.0–49.8)	0.315

CMC: Circulating melanoma cells. Min: Minimum. Max: Maximum. No: Number. * For categorical variables, the first *p*-value corresponds to the result of the Pearson’s Chi-Square test and the second *p*-value in brackets was determined by the Fisher’s exact test. The numerical variables were analyzed by the Mann-Whitney U test. ** Data missing from one patient.

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
