# Peer review of "Monosomy-3 Alters the Expression Profile of the Glucose Transporters GLUT1-3 in Uveal Melanoma"

_ijms, 2020, doi:10.3390/ijms21249345_

Round 1
Reviewer 1 Report
Comments on "Monosomy-3 alters the expression profile of the glucose transporters GLUT1-3 in uveal melanoma"
The manuscript is written clearly and in a comprehensible way.
Glucose transporters play an important role for tumor development, since they are involved in the so called oxidative glycolysis. Supply with glucose is important for tumor growth and progression. A connection between monosomy 3 and glucose transporter de-regulation would contribute to further understanding of uveal melanoma.
Major Points:
-Authors mention GLUT-4 in the introduction and it is clear that is has a prominent role in melanoma progression. Why did authors not investigate GLUT-4 in their samples?
-Authors should cite their own work: "Metastasis of Uveal Melanoma with Monosomy-3 Is Associated with a Less Glycogenetic Gene Expression Profile and the Dysregulation of Glycogen Storage." Cancers (Basel). 2020 Aug; 12(8): 2101.
-Page 4, Row 16: Why are the results from GLUT3 not mentioned?
-Figure 2A and 3A scalebar is missing.
-Figure 3A. Why are the WT samples non pigmented, whereas the monosomy samples show brown pigment? Better pictures should be provided.
Minor Points:
-For some sentences the wording could be even more precise (e.g. Page 1, Row 22: “GLUT3 expression was stronger in the irradiated samples with disomy-3,…” – it is not clear if the expression was stronger compared to native samples or stronger compared to samples with monosomy-3
-Table 1: Abbreviation for “No.” is not explained, but in table 2 it is explained
-Table 1: What does the dot next to “Chromosome 3 status in irradiated tumors, No. (%)” mean?
-Why are there only some p-values written in the graphs? – If only significant p-values are used, why are there non-significant p-values written in figure 1c?
-In figure 2b and 3b the scale bars should be in the same corner.
Author Response
Thank you very much for your kind remarks on our study and your attentive evaluation, which enabled us to improve the clarity of our work.
Please find attached our responses, which describe the changes we have performed according to your recommendations.
Best regards,
Aysegül Tura

Reviewer 2 Report
In this study, the authors aim to analyze the expression of 3 insulin-independent glucose transporters GLUT1, GLUT2, and GLUT3 histochemically and correlate clinical factors from primary tumor samples of uveal melanoma patients. The authors found that metastases were associated with monosomy-3 in the primary tumor cells as well as a higher GLUT1 intensity.
While the premise is interesting and the findings intriguing, this manuscript suffers from some serious flaws both conceptual and technical some of which I have highlighted below. For detailed comments, please check the attached file. This paper needs a major revision because it can be considered for review.
- Results. Not very well described. IHC is to denote % of positive cells. It is not clear how the authors came up with the staining intensity score.
- The sample size is very small and it is a single institute study that should be discussed in the limitations section.
- Some IHC stains have really bad resolution
- Discussion is extremely long and can be cut to one-third of its current size. Many parts of it are rewriting the introduction which is not its purpose.
- The author's claims that "upregulation of the high-affinity glucose transporter GLUT1 is a possible compensation for the low-affinity isoform GLUT2" is not supported in any way by the data since this is a descriptive paper without any mechanistic insights

Author Response
Thank you very much for your thorough evaluation, which drew our attention to the unclear ssues in our manuscript.
Please find attached our responses to your general remarks and the detailed comments.
Best regards,
Aysegül Tura

Round 2
Reviewer 2 Report
I thank the authors for the diligent rebuttal. I appreciate their hard work. I think at this point the authors have answered all my questions. The IHC intensity scoring is not the optimal strategy, but as long as appropriate citations support the materials and methods, it should be alright. To this end I request the authors add references 13 and 14 (from their rebuttal letter) to the materials and methods section where they describe the IHC in addition to reference 15 which they have already added to the manuscript
https://doi.org/10.1186/s13000-014-0221-9
https://doi.org/10.1007/s00262-020-02773-8
This manuscript should be considered for acceptance by the editorial board after this minor revision. I congratulate the authors on this elegant study.
Author Response
Thank you very much for your very kind remarks. We are very pleased to know that we could address your previous comments, which have significantly helped us to improve our manuscript.
In accordance with your minor request, we have inserted the three corresponding citations into the IHC section of our Methods as explained below. The numbers of the subsequent references were corrected accordingly.
Methods, Page 15 (with mark-ups shown), Lines 12-16
"...The semi-quantitative scoring of the immunostainings on these images was performed by an observer who was blinded to the clinical parameters of the patients (T.M.). The scoring method was adapted from earlier procedures that were based on the intensity of the immunoreaction and the area occupied by the cells that exhibit a certain degree of immunolabeling [53,71,72]."
References, Pages 20-22 (with mark-ups shown)
53. Herwig-Carl, M. C.; Sharma, A.; Höller, T.; Holz, F. G.; Schlitter, A. M.; Loeffler, K. U. Spatial Intratumor Heterogeneity in Uveal Melanoma: Tumor Cell Subtypes with a Presumed Invasive Potential Exhibit a Particular Epigenetic Staining Reaction. Exp Eye Res. 2019, 182, 175–181.
71. Fedchenko, N.; Reifenrath, J. Different Approaches for Interpretation and Reporting of Immunohistochemistry Analysis Results in the Bone Tissue - a Review. Diagn. Pathol. 2014, 9, 221.
https://doi.org/10.1186/s13000-014-0221-9
72. Singh, L.; Singh, M. K.; Kenney, M. C.; Jager, M. J.; Rizvi, M. A.; Meel, R.; Lomi, N.; Bakhshi, S.; Sen, S.; Kashyap, S. Prognostic Significance of PD-1/PD-L1 Expression in Uveal Melanoma: Correlation with Tumor-Infiltrating Lymphocytes and Clinicopathological Parameters. Cancer Immunol. Immunother. CII 2020.
https://doi.org/10.1007/s00262-020-02773-8
We thank you very much once more for your time and efforts to attentively evaluate our study.
Kind regards,
Aysegül Tura